# RTFM: Generalising to Novel Environment Dynamics via Reading

**Victor Zhong**[*]
Paul G. Allen School of
Computer Science & Engineering
University of Washington
`vzhong@cs.washington.edu`

**Tim Rocktäschel**
Facebook AI Research &
University College London
`rockt@fb.com`

**Edward Grefenstette**
Facebook AI Research &
University College London
`egrefen@fb.com`

## Abstract

Obtaining policies that can generalise to new environments in reinforcement learning is challenging. In this work, we demonstrate that language understanding via a reading policy learner is a promising vehicle for generalisation to new environments. We propose a grounded policy learning problem, Read to Fight Monsters (RTFM), in which the agent must jointly reason over a language goal, relevant dynamics described in a document, and environment observations. We procedurally generate environment dynamics and corresponding language descriptions of the dynamics, such that agents must read to understand new environment dynamics instead of memorising any particular information. In addition, we propose $\texttt{txt2}\pi$, a model that captures three-way interactions between the goal, document, and observations. On RTFM, $\texttt{txt2}\pi$ generalises to new environments with dynamics not seen during training via reading. Furthermore, our model outperforms baselines such as FiLM and language-conditioned CNNs on RTFM. Through curriculum learning, $\texttt{txt2}\pi$ produces policies that excel on complex RTFM tasks requiring several reasoning and coreference steps.

## 1 Introduction

Reinforcement learning (RL) has been successful in a variety of areas such as continuous control (Lillicrap et al., 2015), dialogue systems (Li et al., 2016), and game-playing (Mnih et al., 2013). However, RL adoption in real-world problems is limited due to poor sample efficiency and failure to generalise to environments even slightly different from those seen during training. We explore language-conditioned policy learning, where agents use machine reading to discover strategies required to solve a task, thereby leveraging language as a means to generalise to new environments.

Prior work on language grounding and language-based RL (see Luketina et al. (2019) for a recent survey) are limited to scenarios in which language specifies the goal for some fixed environment dynamics (Branavan et al., 2011; Hermann et al., 2017; Bahdanau et al., 2019; Fried et al., 2018; Co-Reyes et al., 2019), or the dynamics of the environment vary and are presented in language for some fixed goal (Branavan et al., 2012). In practice, changes to goals and to environment dynamics tend to occur simultaneously—given some goal, we need to find and interpret relevant information to understand how to achieve the goal. That is, the agent should account for variations in both by selectively reading, thereby generalising to environments with dynamics not seen during training.

Our contributions are two-fold. First, we propose a grounded policy learning problem that we call Read to Fight Monsters (RTFM). In RTFM, the agent must jointly reason over a language goal, a document that specifies environment dynamics, and environment observations. In particular, it must identify relevant information in the document to shape its policy and accomplish the goal. To necessitate reading comprehension, we expose the agent to ever changing environment dynamics and corresponding language descriptions such that it cannot avoid reading by memorising any particular environment dynamics. We procedurally generate environment dynamics and natural language templated descriptions of dynamics and goals to produced a combinatorially large number of environment dynamics to train and evaluate RTFM.

---

[*]Work done during an internship at Facebook AI Research.

Second, we propose txt2π to model the joint reasoning problem in RTFM. We show that txt2π generalises to goals and environment dynamics not seen during training, and outperforms previous language-conditioned models such as language-conditioned CNNs and FiLM (Perez et al., 2018; Bahdanau et al., 2019) both in terms of sample efficiency and final win-rate on RTFM. Through curriculum learning where we adapt txt2π trained on simpler tasks to more complex tasks, we obtain agents that generalise to tasks with natural language documents that require five hops of reasoning between the goal, document, and environment observations. Our qualitative analyses show that txt2π attends to parts of the document relevant to the goal and environment observations, and that the resulting agents exhibit complex behaviour such as retrieving correct items, engaging correct enemies after acquiring correct items, and avoiding incorrect enemies. Finally, we highlight the complexity of RTFM in scaling to longer documents, richer dynamics, and natural language variations. We show that significant improvement in language-grounded policy learning is needed to solve these problems in the future.

## 2    RELATED WORK

**Language-conditioned policy learning.**    A growing body of research is learning policies that follow imperative instructions. The granularity of instructions vary from high-level instructions for application control (Branavan, 2012) and games (Hermann et al., 2017; Bahdanau et al., 2019) to step-by-step navigation (Fried et al., 2018). In contrast to learning policies for imperative instructions, Branavan et al. (2011; 2012); Narasimhan et al. (2018) infer a policy for a fixed goal using features extracted from high level strategy descriptions and general information about domain dynamics. Unlike prior work, we study the combination of imperative instructions and descriptions of dynamics. Furthermore, we require that the agent learn to filter out irrelevant information to focus on dynamics relevant to accomplishing the goal.

**Language grounding.**    Language grounding refers to interpreting language in a non-linguistic context. Examples of such context include images (Barnard & Forsyth, 2001), games (Chen & Mooney, 2008; Wang et al., 2016), robot control (Kollar et al., 2010; Tellex et al., 2011), and navigation (Anderson et al., 2018). We study language grounding in interactive games similar to Branavan (2012); Hermann et al. (2017) or Co-Reyes et al. (2019), where executable semantics are not provided and the agent must learn through experience. Unlike prior work, we require grounding between an underspecified goal, a document of environment dynamics, and world observations. In addition, we focus on generalisation to not only new goal descriptions but new environments dynamics.

## 3    READ TO FIGHT MONSTERS

We consider a scenario where the agent must jointly reason over a language **goal**, relevant environment **dynamics** specified in a text document, and **environment observations**. In reading the document, the agent should identify relevant information key to solving the goal in the environment. A successful agent needs to perform this language grounding to generalise to new environments with dynamics not seen during training.

To study generalisation via reading, the environment dynamics must differ every episode such that the agent cannot avoid reading by memorising a limited set of dynamics. Consequently, we procedurally generate a large number of unique environment dynamics (e.g. effective(blessed items, poison monsters)), along with language descriptions of environment dynamics (e.g. blessed items are effective against poison monsters) and goals (e.g. Defeat the order of the forest). We couple a large, customisable ontology inspired by rogue-like games such as NetHack or Diablo, with natural language templates to create a combinatorially rich set of environment dynamics to learn from and evaluate on.

In RTFM, the agent is given a document of environment dynamics, observations of the environment, and an underspecified goal instruction. Figure 1 illustrates an instance of the game. Concretely, we design a set of dynamics that consists of monsters (e.g. wolf, goblin), teams (e.g. Order of the Forest), element types (e.g. fire, poison), item modifiers (e.g. fanatical, arcane), and items (e.g. sword, hammer). When the player is in the same cell with a monster or weapon, the player picks up the item or engages in combat with the monster. The player can possess one item at a time, and drops existing

**Doc:**
The Rebel Enclave consists of jackal, spider, and warg. Arcane, blessed items are useful for poison monsters. Star Alliance contains bat, panther, and wolf. Goblin, jaguar, and lynx are on the same team - they are in the Order of the Forest. Gleaming and mysterious weapons beat cold monsters. Lightning monsters are weak against Grandmaster's and Soldier's weapons. Fire monsters are defeated by fanatical and shimmering weapons.

**Goal:**
Defeat the Order of the Forest

Figure 1: RTFM requires jointly reasoning over the goal, a document describing environment dynamics, and environment observations. This figure shows key snapshots from a trained policy on one randomly sampled environment. Frame 1 shows the initial world. In 4, the agent approaches "fanatical sword", which beats the target "fire goblin". In 5, the agent acquires the sword. In 10, the agent evades the distractor "poison bat" while chasing the target. In 11, the agent engages the target and defeats it, thereby winning the episode. Sprites are used for visualisation — the agent observes cell content in text (shown in white). More examples are in appendix A.

weapons if they pick up a new weapon. A monster moves towards the player with 60% probability, and otherwise moves randomly. The dynamics, the agent's inventory, and the underspecified goal are rendered as text. The game world is rendered as a matrix of text in which each cell describes the entity occupying the cell. We use human-written templates for stating which monsters belong to which team, which modifiers are effective against which element, and which team the agent should defeat (see appendix H for details on collection and G for a list of entities in the game). In order to achieve the goal, the agent must cross-reference relevant information in the document and as well as in the observations.

During every episode, we subsample a set of groups, monsters, modifiers, and elements to use. We randomly generate group assignments of which monsters belong to which team and which modifier is effective against which element. A document that consists of randomly ordered statements corresponding to this group assignment is presented to the agent. We sample one element, one team, and a monster from that team (e.g. "fire goblin" from "Order of the forest") to be the target monster. Additionally, we sample one modifier that beats the element and an item to be the item that defeats the target monster (e.g. "fanatical sword"). Similarly, we sample an element, a team, and a monster from a different team to be the distractor monster (e.g. poison bat), as well as an item that defeats the distractor monster (e.g. arcane hammer).

In order to win the game (e.g. Figure 1), the agent must

1. identify the target team from the goal (e.g. Order of the Forest)
2. identify the monsters that belong to that team (e.g. goblin, jaguar, and ghost)
3. identify which monster is in the world (e.g. goblin), and its element (e.g. fire)
4. identify the modifiers that are effective against this element (e.g. fanatical, shimmering)
5. find which modifier is present (e.g. fanatical), and the item with the modifier (e.g. sword)

    6. pick up the correct item (e.g. fanatical sword)

    7. engage the correct monster in combat (e.g. fire goblin).

If the agent deviates from this trajectory (e.g. does not have correct item before engaging in combat, engages with distractor monster), it cannot defeat the target monster and therefore will lose the game. The agent receives a reward of +1 if it wins the game and -1 otherwise.

RTFM presents challenges not found in prior work in that it requires a large number of grounding steps in order to solve a task. In order to perform this grounding, the agent must jointly reason over a language goal and document of dynamics, as well as environment observations. In addition to the environment, the positions of the target and distractor within the document are randomised—the agent cannot memorise ordering patterns in order to solve the grounding problems, and must instead identify information relevant to the goal and environment at hand.

We split environments into train and eval sets. No assignments of monster-team-modifier-element are shared between train and eval to test whether the agent is able to generalise to new environments with dynamics not seen during training via reading. There are more than 2 million train or eval environments without considering the natural language templates, and 200 million otherwise. With random ordering of templates, the number of unique documents exceeds 15 billion.

## 4 MODEL

We propose the `txt2π` model, which builds representations that capture three-way interactions between the goal, document describing environment dynamics, and environment observations. We begin with definition of the Bidirectional Feature-wise Linear Modulation (FiLM$^2$) layer, which forms the core of our model.

### 4.1 BIDIRECTIONAL FEATURE-WISE LINEAR MODULATION (FiLM$^2$) LAYER

Feature-wise linear modulation (FiLM), which modulates visual inputs using representations of textual instructions, is an effective method for image captioning (Perez et al., 2018) and instruction following (Bahdanau et al., 2019). In RTFM, the agent must not only filter concepts in the visual domain using language but filter concepts in the text domain using visual observations. To support this, FiLM$^2$ builds

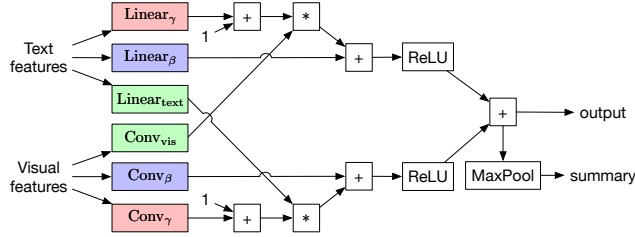

Figure 2: The FiLM$^2$ layer.

codependent representations of text and visual inputs by further incorporating conditional representations of the text given visual observations. Figure 2 shows the FiLM$^2$ layer.

We use upper-case bold letters to denote tensors, lower-case bold letters for vectors, and non-bold letters for scalars. Exact dimensions of these variables are shown in Table 4 in appendix B. Let $\boldsymbol{x}_{\text{text}}$ denote a fixed-length $d_{\text{text}}$-dimensional representation of the text and $\boldsymbol{X}_{\text{vis}}$ the representation of visual inputs with height $H$, width $W$, and $d_{\text{vis}}$ channels. Let Conv denote a convolution layer. Let + and * symbols denote element-wise addition and multiplication operations that broadcast over spatial dimensions. We first modulate visual features using text features:

$$
\begin{aligned}
\boldsymbol{\gamma}_{\text{text}} &= \boldsymbol{W}_\gamma \boldsymbol{x}_{\text{text}} + \boldsymbol{b}_\gamma & (1) \\
\boldsymbol{\beta}_{\text{text}} &= \boldsymbol{W}_\beta \boldsymbol{x}_{\text{text}} + \boldsymbol{b}_\beta & (2) \\
\boldsymbol{V}_{\text{vis}} &= \text{ReLU}((1 + \boldsymbol{\gamma}_{\text{text}}) * \text{Conv}_{\text{vis}}(\boldsymbol{X}_{\text{vis}}) + \boldsymbol{\beta}_{\text{text}}) & (3)
\end{aligned}
$$

Unlike FiLM, we additionally modulate text features using visual features:

$$
\begin{aligned}
\boldsymbol{\Gamma}_{\text{vis}} &= \text{Conv}_\gamma(\boldsymbol{X}_{\text{vis}}) & (4) \\
\boldsymbol{B}_{\text{vis}} &= \text{Conv}_\beta(\boldsymbol{X}_{\text{vis}}) & (5) \\
\boldsymbol{V}_{\text{text}} &= \text{ReLU}((1 + \boldsymbol{\Gamma}_{\text{vis}}) * (\boldsymbol{W}_{\text{text}} \boldsymbol{x}_{\text{text}} + \boldsymbol{b}_{\text{text}}) + \boldsymbol{B}_{\text{vis}}) & (6)
\end{aligned}
$$

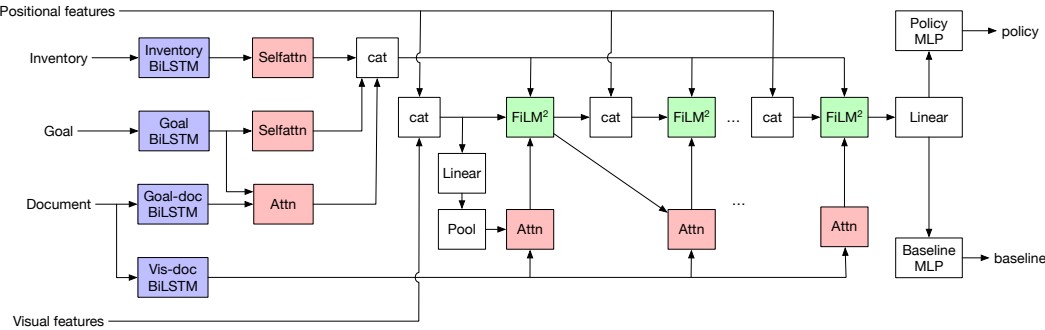

Figure 3: txt2$\pi$ models interactions between the goal, document, and observations.

The output of the FiLM$^2$ layer consists of the sum of the modulated features $\boldsymbol{V}$, as well as a max-pooled summary $\boldsymbol{s}$ over this sum across spatial dimensions.

$$\boldsymbol{V} = \boldsymbol{V}_{\text{vis}} + \boldsymbol{V}_{\text{text}} \qquad (7) \qquad\qquad \boldsymbol{s} = \text{MaxPool}(\boldsymbol{V}) \qquad (8)$$

## 4.2 The txt2$\pi$ model

We model interactions between observations from the environment, goal, and document using FiLM$^2$ layers. We first encode text inputs using bidirectional LSTMs, then compute summaries using self-attention and conditional summaries using attention. We concatenate text summaries into text features, which, along with visual features, are processed through consecutive FiLM$^2$ layers. In this case of a textual environment, we consider the grid of word embeddings as the visual features for FiLM$^2$. The final FiLM$^2$ output is further processed by MLPs to compute a policy distribution over actions and a baseline for advantage estimation. Figure 3 shows the txt2$\pi$ model.

Let $\boldsymbol{E}_{\text{obs}}$ denote word embeddings corresponding to the observations from the environment, where $\boldsymbol{E}_{\text{obs}}[:,:,i,j]$ represents the embeddings corresponding to the $l_{\text{obs}}$-word string that describes the objects in location $(i, j)$ in the grid-world. Let $\boldsymbol{E}_{\text{doc}}$, $\boldsymbol{E}_{\text{inv}}$, and $\boldsymbol{E}_{\text{goal}}$ respectively denote the embeddings corresponding to the $l_{\text{doc}}$-word document, the $l_{\text{inv}}$-word inventory, and the $l_{\text{goal}}$-word goal. We first compute a fixed-length summary $\boldsymbol{c}_{\text{goal}}$ of the the goal using a bidirectional LSTM (Hochreiter & Schmidhuber, 1997) followed by self-attention (Lee et al., 2017; Zhong et al., 2018).

$$\boldsymbol{H}_{\text{goal}} = \text{BiLSTM}_{\text{goal}}(\boldsymbol{E}_{\text{goal}}) \qquad (9) \qquad\qquad a'_{\text{goal},i} = \boldsymbol{w}_{\text{goal}}\boldsymbol{h}^{\mathsf{T}}_{\text{goal},i} + b_{\text{goal}} \qquad (10)$$

$$\boldsymbol{a}_{\text{goal}} = \text{softmax}(\boldsymbol{a}'_{\text{goal}}) \qquad (11) \qquad\qquad \boldsymbol{c}_{\text{goal}} = \sum_{i=1}^{l_{\text{goal}}} a_{\text{goal},i}\boldsymbol{h}_{\text{goal},i} \qquad (12)$$

We abbreviate self-attention over the goal as $\boldsymbol{c}_{\text{goal}} = \text{selfattn}(\boldsymbol{H}_{\text{goal}})$. We similarly compute a summary of the inventory as $\boldsymbol{c}_{\text{inv}} = \text{selfattn}(\text{BiLSTM}_{\text{inv}}(\boldsymbol{E}_{\text{inv}}))$. Next, we represent the document encoding conditioned on the goal using dot-product attention (Luong et al., 2015).

$$\boldsymbol{H}_{\text{doc}} = \text{BiLSTM}_{\text{goal-doc}}(\boldsymbol{E}_{\text{doc}}) \qquad (13) \qquad\qquad a'_{\text{doc},i} = \boldsymbol{c}_{\text{goal}}\boldsymbol{h}^{\mathsf{T}}_{\text{doc},i} \qquad (14)$$

$$\boldsymbol{a}_{\text{doc}} = \text{softmax}(\boldsymbol{a}'_{\text{doc}}) \qquad (15) \qquad\qquad \boldsymbol{c}_{\text{doc}} = \sum_{i=1}^{l_{\text{doc}}} a_{\text{doc},i}\boldsymbol{h}_{\text{doc},i} \qquad (16)$$

We abbreviate attention over the document encoding conditioned on the goal summary as $\boldsymbol{c}_{\text{doc}} = \text{attend}(\boldsymbol{H}_{\text{doc}}, \boldsymbol{c}_{\text{goal}})$. Next, we build the joint representation of the inputs using successive FiLM$^2$ layers. At each layer, the visual input to the FiLM$^2$ layer is the concatenation of the output of the previous layer with positional features. For each cell, the positional feature $\boldsymbol{X}_{\text{pos}}$ consists of the $x$ and $y$ distance from the cell to the agent's position respectively, normalized by the width and height of the grid-world. The text input is the concatenation of the goal summary, the inventory summary, the attention over the document given the goal, and the attention over the document given the previous visual summary. Let $[a; b]$ denote the feature-wise concatenation of $a$

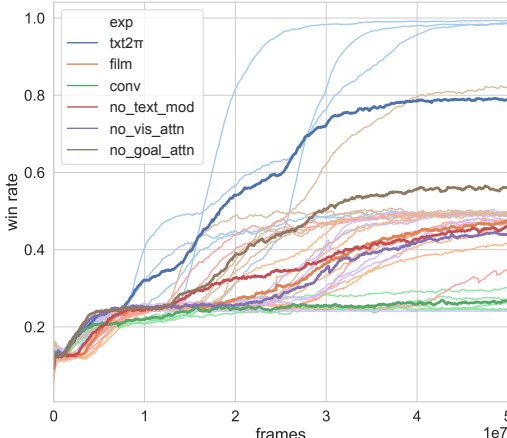

| Model | Win rate | | |
|---|---|---|---|
| | Train | Eval $6\times6$ | Eval $10\times10$ |
| conv | $24\pm0$ | $25\pm1$ | $13\pm1$ |
| FiLM | $49\pm1$ | $49\pm2$ | $32\pm3$ |
| no_task_attn | $49\pm2$ | $49\pm2$ | $35\pm6$ |
| no_vis_attn | $49\pm2$ | $49\pm1$ | $40\pm12$ |
| no_text_mod | $49\pm1$ | $49\pm2$ | $35\pm2$ |
| txt2$\pi$ | $84\pm21$ | $83\pm21$ | $66\pm22$ |

Table 1: Final win rate on simplest variant of RTFM. The models are trained on one set of dynamics (e.g. training set) and evaluated on another set of dynamics (e.g. evaluation set). "Train" and "Eval" show final win rates on training and eval environments.

Figure 4: Ablation training curves on simplest variant of RTFM. Individual runs are in light colours. Average win rates are in bold, dark lines.

and $b$. For the $i$th layer, we have

$$R^{(i)} = [V^{(i-1)}; X_{\text{pos}}] \tag{17}$$

$$T^{(i)} = [c_{\text{goal}}; c_{\text{inv}}; c_{\text{doc}}; \text{attend}(\text{BiLSTM}_{\text{vis-doc}}(E_{\text{doc}}), s^{(i-1)})] \tag{18}$$

$$V^{(i)}, s^{(i)} = \text{FiLM}^{2(i)}(R^{(i)}, T^{(i)}) \tag{19}$$

$\text{BiLSTM}_{\text{vis-doc}}(E_{\text{doc}})$ is another encoding of the document similar to $H_{\text{goal}}$, produced using a separate LSTM, such that the document is encoded differently for attention with the visual features and with the goal. For $i = 0$, we concatenate the bag-of-words embeddings of the grid with positional features as the initial visual features $V^{(0)} = [\sum_j E_{\text{obs},j}; X_{\text{pos}}]$. We max pool a linear transform of the initial visual features to compute the initial visual summary $s^{(0)} = \text{MaxPool}(W_{\text{ini}}V^{(0)} + b_{\text{ini}})$. Let $s^{(\text{last})}$ denote visual summary of the last FiLM$^2$ layer. We compute the policy $y_{\text{policy}}$ and baseline $y_{\text{baseline}}$ as

$$o = \text{ReLU}(W_o s^{(\text{last})} + b_o) \tag{20}$$

$$y_{\text{policy}} = \text{MLP}_{\text{policy}}(o) \tag{21}$$

$$y_{\text{baseline}} = \text{MLP}_{\text{baseline}}(o) \tag{22}$$

where $\text{MLP}_{\text{policy}}$ and $\text{MLP}_{\text{baseline}}$ are 2-layer multi-layer perceptrons with ReLU activation. We train using TorchBeast (Küttler et al., 2019), an implementation of IMPALA (Espeholt et al., 2018). Please refer to appendix D for details.

## 5 EXPERIMENTS

We consider variants of RTFM by varying the size of the grid-world ($6 \times 6$ vs $10 \times 10$), allowing many-to-one group assignments to make disambiguation more difficult (group), allowing dynamic, moving monsters that hunt down the player (dyna), and using natural language templated documents (nl). In the absence of many-to-one assignments, the agent does not need to perform steps 3 and 5 in section 3 as there is no need to disambiguate among many assignees, making it easier to identify relevant information.

We compare txt2$\pi$ to the FiLM model by Bahdanau et al. (2019) and a language-conditioned residual CNN model. We train on one set of dynamics (e.g. group assignments of monsters and modifiers) and evaluated on a held-out set of dynamics. We also study three variants of txt2$\pi$. In no_task_attn, the document attention conditioned on the goal utterance (equation 16) is removed and the goal instead represented through self-attention and concatenated with the rest of the text features. In no_vis_attn, we do not attend over the document given the visual output of the previous layer (equation 18), and the document is instead represented through self-attention.

| Transfer from | Transfer to | | | | | | | |
|---|---|---|---|---|---|---|---|---|
| | $6 \times 6$ | $6 \times 6$ dyna | $6 \times 6$ groups | $6 \times 6$ nl | $6 \times 6$ dyna groups | $6 \times 6$ group nl | $6 \times 6$ dyna nl | $6 \times 6$ dyna group nl |
| random | $\mathbf{84 \pm 20}$ | $26 \pm 7$ | $25 \pm 3$ | $45 \pm 6$ | $23 \pm 2$ | $25 \pm 3$ | $23 \pm 2$ | $23 \pm 2$ |
| +$6 \times 6$ | | $\mathbf{85 \pm 9}$ | $82 \pm 19$ | $78 \pm 24$ | $64 \pm 12$ | $52 \pm 13$ | $53 \pm 18$ | $40 \pm 8$ |
| +dyna | | | | | $\mathbf{77 \pm 10}$ | | $65 \pm 16$ | $43 \pm 4$ |
| +group | | | | | | | | $\mathbf{65 \pm 17}$ |

Table 2: Curriculum training results. We keep 5 randomly initialised models through the entire curriculum. A cell in row $i$ and column $j$ shows transfer from the best-performing setting in the previous stage (bold in row $i - 1$) to the new setting in column $j$. Each cell shows final mean and standard deviation of win rate on the training environments. Each experiment trains for 50 million frames, except for the initial stage (first row, 100 million instead). For the last stage (row 4), we also transfer to a $10 \times 10 + \text{dyna} + \text{group} + \text{nl}$ variant and obtain $\mathbf{61 \pm 18}$ win rate.

In no_text_mod, text modulation using visual features (equation 6) is removed. Please see appendix C for model details on our model and baselines, and appendix D for training details.

## 5.1 COMPARISON TO BASELINES AND ABLATIONS

We compare txt2$\pi$ to baselines and ablated variants on a simplified variant of RTFM in which there are one-to-one group assignments (no group), stationary monsters (no dyna), and no natural language templated descriptions (no nl). Figure 4 shows that compared to baselines and ablated variants, txt2$\pi$ is more sample efficient and converges to higher performance. Moreover, no ablated variant is able to solve the tasks—it is the combination of ablated features that enables txt2$\pi$ to win consistently. Qualitatively, the ablated variants converge to locally optimum policies in which the agent often picks up a random item and then attacks the correct monster, resulting in a $\sim 50\%$ win rate. Table 1 shows that all models, with the exception of the CNN baseline, generalise to new evaluation environments with dynamics and world configurations not seen during training, with txt2$\pi$ outperforming FiLM and the CNN model.

We find similar results for txt2$\pi$, its ablated variants, and baselines on a separate, language-based rock-paper-scissors task in which the agent needs to deduce cyclic dependencies (which type beats which other type) through reading in order to acquire the correct item and defeat a monster. We observe that the performance of reading models transfer from training environments to new environments with unseen types and unseen dependencies. Compared to ablated variants and baselines, txt2$\pi$ is more sample efficient and achieves higher performance on both training and new environment dynamics. When transferring to new environments, txt2$\pi$ remains more sample efficient than the other models. Details on these experiments are found in appendix E.

## 5.2 CURRICULUM LEARNING FOR COMPLEX ENVIRONMENTS

Due to the long sequence of co-references the agent must perform in order to solve the full RTFM ($10 \times 10$ with moving monsters, many-to-one group assignments, and natural language templated documents) we design a curriculum to facilitate policy learning by starting with simpler variants of RTFM. We start with the simplest variant (no group, no dyna, no nl) and then add in an additional dimension of complexity. We repeatedly add more complexity until we obtain $10 \times 10$ worlds with moving monsters, many-to-one group assignments and natural language templated descriptions. The performance across the curriculum is shown in Table 2

| Train env | Eval env | Win rate | |
|---|---|---|---|
| | | Train | Eval |
| $6 \times 6$ | $6 \times 6$ | $65 \pm 17$ | $55 \pm 22$ |
| | $10 \times 10$ | | $55 \pm 27$ |
| $10 \times 10$ | $10 \times 10$ | $61 \pm 18$ | $43 \pm 13$ |

Table 3: Win rate when evaluating on new dynamics and world configurations for txt2$\pi$ on the full RTFM problem.

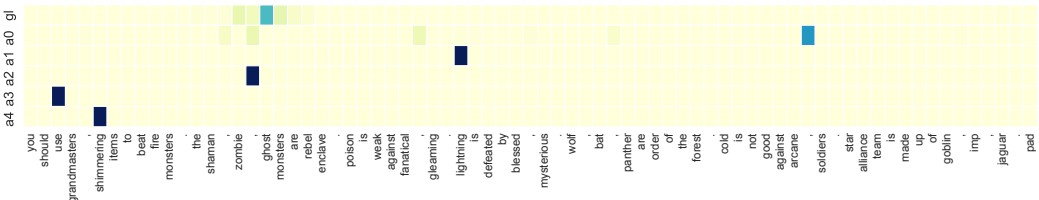

(a) The entities present are shimmering morning star, mysterious spear, fire jaguar, and lightning ghost.

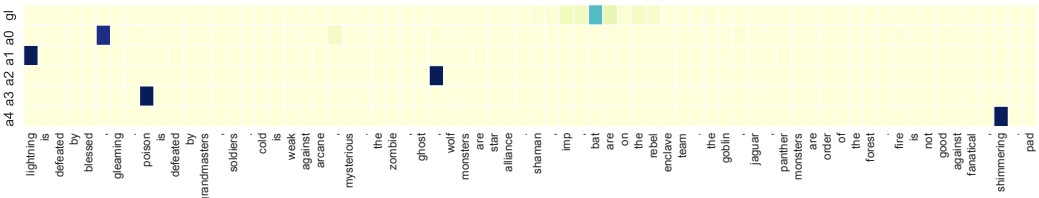

(b) The entities present are soldier's axe, shimmering axe, fire shaman, and poison wolf.

Figure 5: txt2$\pi$ attention on the full RTFM. These include the document attention conditioned on the goal (top) as well as those conditioned on summaries produced by intermediate FiLM$^2$ layers. Weights are normalised across words (e.g. horizontally). Darker means higher attention weight.

(see Figure 13 in appendix F for training curves of each stage). We see that curriculum learning is crucial to making progress on RTFM, and that initial policy training (first row of Table 2) with additional complexities in any of the dimensions result in significantly worse performance. We take each of the 5 runs after training through the whole curriculum and evaluate them on dynamics not seen during training. Table 3 shows variants of the last stage of the curriculum in which the model was trained on $6 \times 6$ versions of the full RTFM and in which the model was trained on $10 \times 10$ versions of the full RTFM. We see that models trained on smaller worlds generalise to bigger worlds. Despite curriculum learning, however, performance of the final model trail that of human players, who can consistently solve RTFM. This highlights the difficulties of the RTFM problem and suggests that there is significant room for improvement in developing better language grounded policy learners.

**Attention maps.**    Figure 5 shows attention conditioned on the goal and on observation summaries produced by intermediate FiLM$^2$ layers. Goal-conditioned attention consistently locates the clause that contains the team the agent is supposed to attack. Intermediate layer attentions focus on regions near modifiers and monsters, particularly those that are present in the observations. These results suggests that attention mechanisms in txt2$\pi$ help identify relevant information in the document.

**Analysis of trajectories and failure modes.**    We examine trajectories from well-performing policies (80% win rate) as well as poorly-performing policies (50% win rate) on the full RTFM. We find that well-performing policies exhibit a number of consistent behaviours such as identifying the correct item to pick up to fight the target monster, avoiding distractors, and engaging target monsters after acquiring the correct item. In contrast, the poorly-performing policies occasionally pick up the wrong item, causing the agent to lose when engaging with a monster. In addition, it occasionally gets stuck in evading monsters indefinitely, causing the agent to lose when the time runs out. Replays of both policies can be found in GIFs in the supplementary materials[1].

## 6    CONCLUSION

We proposed RTFM, a grounded policy learning problem in which the agent must jointly reason over a language goal, relevant dynamics specified in a document, and environment observations. In order to study RTFM, we procedurally generated a combinatorially large number of environment dynamics such that the model cannot memorise a set of environment dynamics and must instead generalise via reading. We proposed txt2$\pi$, a model that captures three-way interactions between the goal,

---

[1]Trajectories by txt2$\pi$ on RTFM can be found at https://gofile.io/?c=9k7ZLk

document, and observations, and that generalises to new environments with dynamics not seen during training. txt2π outperforms baselines such as FiLM and language-conditioned CNNs. Through curriculum learning, txt2π performs well on complex RTFM tasks that require several reasoning and coreference steps with natural language templated goals and descriptions of the dynamics. Our work suggests that language understanding via reading is a promising way to learn policies that generalise to new environments. Despite curriculum learning, our best models trail performance of human players, suggesting that there is ample room for improvement in grounded policy learning on complex RTFM problems. In addition to jointly learning policies based on external documentation and language goals, we are interested in exploring how to use supporting evidence in external documentation to reason about plans (Andreas et al., 2018) and induce hierarchical policies (Hu et al., 2019; Jiang et al., 2019).

## ACKNOWLEDGEMENT

We thank Heinrich Küttler and Nantas Nardelli for their help in adapting TorchBeast and the FAIR London team for their feedback and support.

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

## A  PLAYTHROUGH EXAMPLES

These figures shows key snapshots from a trained policy on randomly sampled environments.

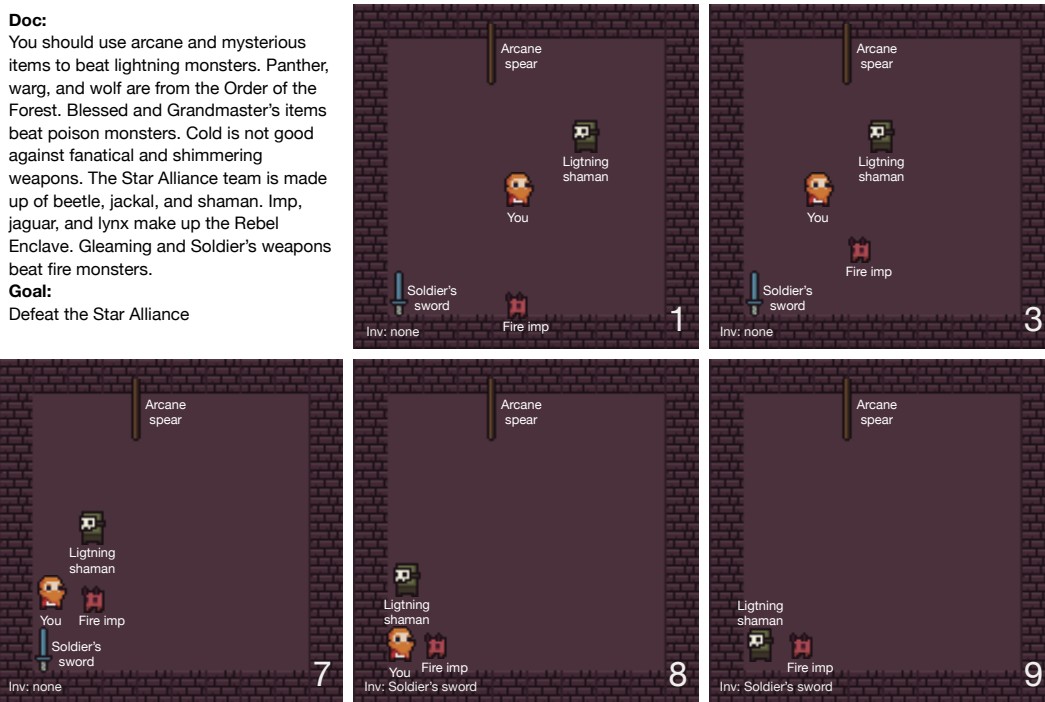

Figure 6: The initial world is shown in 1. In 4, the agent avoids the target "lightning shaman" because it does not yet have "arcane spear", which beats the target. In 7 and 8, the agent is cornered by monsters. In 9, the agent is forced to engage in combat and loses.

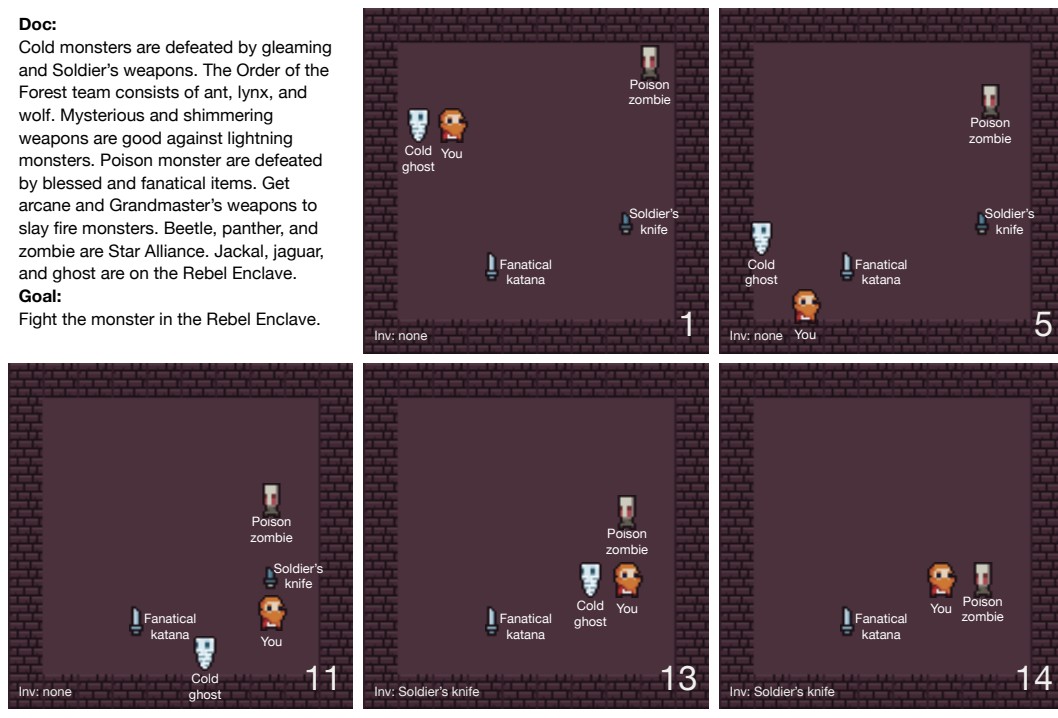

Figure 7: The initial world is shown in 1. In 5 the agent evades the target "cold ghost" because it does not yet have "soldier's knife", which beats the target. In 11 and 13, the agent obtains "soldier's knife" while evading monsters. In 14, the agent defeats the target and wins.

## B    VARIABLE DIMENSIONS

Let $\boldsymbol{x}_{\text{text}} \in \mathbb{R}^{d_{\text{text}}}$ denote a fixed-length $d_{\text{text}}$-dimensional representation of the text and $\boldsymbol{X}_{\text{vis}} \in \mathbb{R}^{d_{\text{vis}} \times H \times W}$ denote the representation of visual inputs with

| Variable | Symbol | Dimension |
|---|---|---|
| $d_{\text{text}}$-dim text representation | $\boldsymbol{x}_{\text{text}}$ | $d_{\text{text}}$ |
| $d_{\text{vis}}$-dim visual representation with height $H$, width $W$, $d_{\text{vis}}$ channels | $\boldsymbol{X}_{\text{vis}}$ | $d_{\text{vis}} \times H \times W$ |
| Environment observations embeddings | $\boldsymbol{E}_{\text{obs}}$ | $l_{\text{obs}} \times d_{\text{emb}} \times H \times W$ |
| $l_{\text{obs}}$-word string that describes the objects in location $(i, j)$ in the grid-world | $\boldsymbol{E}_{\text{obs}}[:, :, i, j]$ | $l_{\text{obs}} \times d_{\text{emb}}$ |
| $l_{\text{doc}}$-word document embeddings | $\boldsymbol{E}_{\text{doc}}$ | $l_{\text{doc}} \times d_{\text{emb}}$ |
| $l_{\text{inv}}$-word inventory embeddings | $\boldsymbol{E}_{\text{inv}}$ | $l_{\text{inv}} \times d_{\text{emb}}$ |
| $l_{\text{goal}}$-word goal embeddings | $\boldsymbol{E}_{\text{goal}}$ | $l_{\text{goal}} \times d_{\text{emb}}$ |

Table 4: Variable dimensions

## C  MODEL DETAILS

### C.1  TXT2π

**Hyperparameters.**  The txt2π used in our experiments consists of 5 consecutive FiLM[2] layers, each with 3x3 convolutions and padding and stride sizes of 1. The txt2π layers have channels of 16, 32, 64, 64, and 64, with residual connections from the 3rd layer to the 5th layer. The Goal-doc LSTM (see Figure 3) shares weight with the Goal LSTM. The Inventory and Goal LSTMs have a hidden dimension of size 10, whereas the Vis-doc LSTM has a dimension of 100. We use a word embedding dimension of 30.

### C.2  CNN WITH RESIDUAL CONNECTIONS

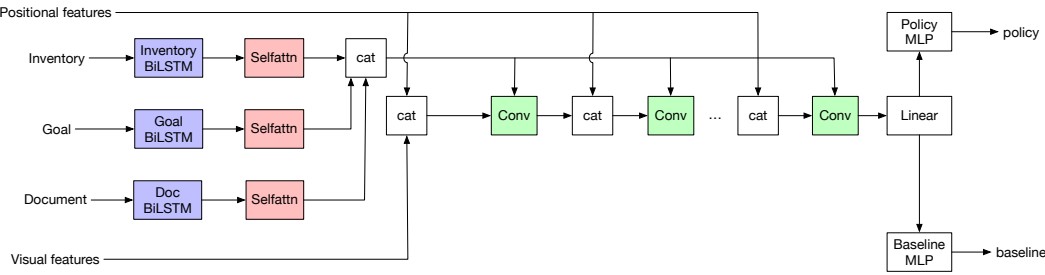

Figure 8: The convolutional network baseline. The FiLM baseline has the same structure, but with convolutional layers replaced by FiLM layers.

Like txt2π, the CNN baseline consists of 5 layers of convolutions with channels of 16, 32, 64, 64, and 64. There are residual connections from the 3rd layer to the 5th layer. The input to each layer consists of the output of the previous layer, concatenated with positional features.

The input to the network is the concatenation of the observations $V^{(0)}$ and text representations. The text representations consist of self-attention over bidirectional LSTM-encoded goal, document, and inventory. These attention outputs are replicated over the dimensions of the grid and concatenated feature-wise with the observation embeddings in each cell. Figure 8 illustrates the CNN baseline.

### C.3  FILM BASELINE

The FiLM baseline encodes text in the same fashion as the CNN model. However, instead of using convolutional layers, each layer is a FiLM layer from Bahdanau et al. (2019). Note that in our case, the language representation is a self-attention over the LSTM states instead of a concatenation of terminal LSTM states.

## D  TRAINING PROCEDURE

We train using an implementation of IMPALA (Espeholt et al., 2018). In particular, we use 20 actors and a batch size of 24. When unrolling actors, we use a maximum unroll length of 80 frames. Each episode lasts for a maximum of 1000 frames. We optimise using RMSProp (Tieleman & Hinton, 2012) with a learning rate of 0.005, which is annealed linearly for 100 million frames. We set $\alpha = 0.99$ and $\epsilon = 0.01$.

During training, we apply a small negative reward for each time step of $-0.02$ and a discount factor of 0.99 to facilitate convergence. We additionally include a entropy cost to encourage exploration. Let $\boldsymbol{y}_{\text{policy}}$ denote the policy. The entropy loss is calculated as

$$L_{\text{policy}} = -\sum_i \boldsymbol{y}_{\text{policy}_i} \log \boldsymbol{y}_{\text{policy}_i} \tag{23}$$

In addition to policy gradient, we add in the entropy loss with a weight of 0.005 and the baseline loss with a weight of 0.5. The baseline loss is computed as the root mean square of the advantages (Espeholt et al., 2018).

When tuning models, we perform a grid search using the training environments to select hyperparameters for each model. We train 5 runs for each configuration in order to report the mean and standard deviation. When transferring, we transfer each of the 5 runs to the new task and once again report the mean and standard deviation.

| Scenario | # graphs | | | # edges | | | # nodes | | |
|---|---|---|---|---|---|---|---|---|---|
| | train | dev | unseen | train | dev | % new | train | dev | % new |
| permutation | 30 | 30 | y | 20 | 20 | n | 60 | 60 | n |
| new edge | 20 | 20 | y | 48 | 36 | y | 17 | 13 | n |
| new edge+nodes | 60 | 60 | y | 20 | 20 | y | 5 | 5 | y |

Table 5: Statistics of the three variations of the Rock-paper-scissors task

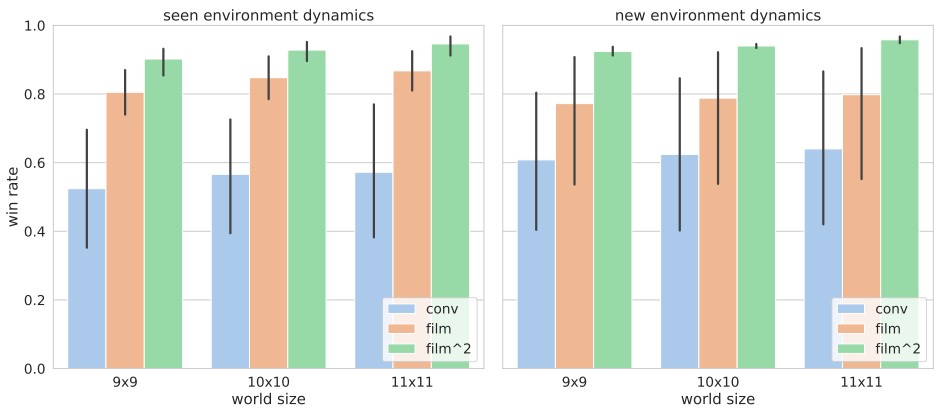

Figure 10: Performance on the Rock-paper-scissors task across models. Left shows final performance on environments whose goals and dynamics were seen during training. Right shows performance on the environments whose goals and dynamics were not seen during training.

## E  ROCK-PAPER-SCISSORS

In addition to the main RTFM tasks, we also study a simpler formulation called Rock-paper-scissors that has a fixed goal. In Rock-paper-scissors, the agent must interpret a document that describes the environment dynamics in order to solve the task. Given an set of characters (e.g. a-z), we sample 3 characters and set up a rock-paper-scissors-like dependency graph between the characters (e.g. "a beats b, b beats c, c beats a"). We then spawn a monster in the world with a randomly assigned type (e.g. "b goblin"), as well as an item corresponding to each type (e.g. "a", "b", and "c"). The attributes of the agent, monster, and items are set up such that the player must obtain the correct item and then engage the monster in order to win. Any other sequence of actions (e.g. engaging the monster without the correct weapon) results in a loss.

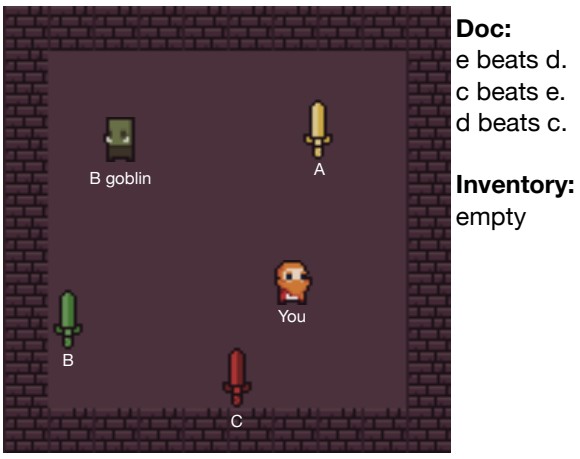

Figure 9: The Rock-paper-scissors task requires jointly reasoning over the game observations and a document describing environment dynamics. The agent observes cell content in the form of text (shown in white).

The winning policy should then be to first identify the type of monster present, then cross-reference the document to find which item defeats that type, then pick up the item, and finally engage the monster in combat. Figure 9 shows an instance of Rock-paper-scissors.

**Reading models generalise to new environments.** We split environment dynamics by permuting 3-character dependency graphs from an alphabet, which we randomly split into training and held-out sets. This corresponds to the "permutations" setting in Table 5.

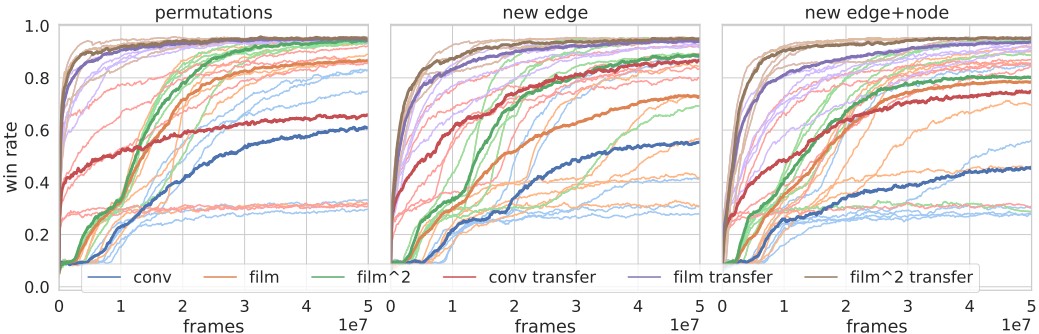

Figure 11: Learning curve while transferring to the development environments. Win rates of individual runs are shown in light colours. Average win rates are shown in bold, dark lines.

We train models on the $10 \times 10$ worlds from the training set and evaluate them on both seen and not seen during training. The left of Figure 10 shows the performance of models on worlds of varying sizes with training environment dynamics. In this case, the dynamics (e.g. dependency graphs) were seen during training. For $9 \times 9$ and $11 \times 11$ worlds, the world configuration not seen during training. For $10 \times 10$ worlds, there is a 5% chance that the initial frame was seen during training.[2] Figure 10 shows the performance on held-out environments not seen during training. We see that all models generalise to environments not seen during training, both when the world configuration is not seen (left) and when the environment dynamics are not seen (right).

**Reading models generalise to new concepts.** In addition to splitting via permutations, we de-

Figure 12: Ablation training curves. Win rates of individual runs are shown in light colours. Average win rates are shown in bold, dark lines.

vise two additional ways of splitting environment dynamics by introducing new edges and nodes into the held-out set. Table 5 shows the three different settings. For each, we study the transfer behaviour of models on new environments. Figure 11 shows the learning curve when training a model on the held-out environments directly and when transferring the model trained on train environments to held-out environments. We observe that all models are significantly more sample-efficient when transferring from training environments, despite the introduction of new edges and new nodes.

**txt2$\pi$ is more sample-efficient and learns better policies.** In Figure 10, we see that the FiLM model outperforms the CNN model on both training environment dynamics and held-out environment dynamics. txt2$\pi$ further outperforms FiLM, and does so more consistently in that the final performance has less variance. This behaviour is also observed in the in Figure 11. When training on the held-out set without transferring, txt2$\pi$ is more sample efficient than FiLM and the CNN model, and achieves higher win-rate. When transferring to the held-out set, txt2$\pi$ remains more sample efficient than the other models.

---

[2]There are 24360 unique grid configurations given a particular dependency graph, 4060 unique dependency graphs in the training set, and 50 million frames seen during training. After training, the model finishes an episode in approximately 10 frames. Hence the probability of seeing a redundant initial frame is $\frac{5e7/10}{24360*4060} = 5\%$.

## F   CURRICULUM LEARNING TRAINING CURVES

Figure 13: Curriculum learning results for txt2$\pi$ on RTFM. Win rates of individual runs are shown in light colours. Average win rates are shown in bold, dark lines.

## G   ENTITIES AND MODIFIERS

Below is a list of entities and modifiers contained in RTFM:

Monsters: wolf, jaguar, panther, goblin, bat, imp, shaman, ghost, zombie

Weapons: sword, axe, morningstar, polearm, knife, katana, cutlass, spear

Elements: cold, fire, lightning, poison

Modifiers: Grandmaster's, blessed, shimmering, gleaming, fanatical, mysterious, Soldier's, arcane

Teams: Star Alliance, Order of the Forest, Rebel Enclave

## H   LANGUAGE TEMPLATES

We collect human-written natural language templates for the goal and the dynamics. The goal statements in RTFM describe which team the agent should defeat. We collect 12 language templates for goal statements. The document of environment dynamics consists of two types of statements. The first type describes which monsters are assigned to with team. The second type describes which modifiers, which describe items, are effective against which element types, which are associated with monsters. We collection 10 language templates for each type of statements. The entire document is composed from statements, which are randomly shuffled. We randomly sample a template for each statement, which we fill with the monsters and team for the first type and modifiers and element for the second type.

