# OpenReview forum: "RTFM: Generalising to New Environment Dynamics via Reading"
_ICLR.cc/2020/Conference — Accept (Poster)_

### Official Review · AnonReviewer2 · 2019-10-23
**Official Blind Review #2**

**Rating:** 6

**Review:**

This work proposes a new environment, Read to Fight Monsters (RTFM), and correspondingly a new algorithm, txt2\pi, for solving this problem. The RTFM requires the agent to read a description of the rules (x beats y, etc) and a description of the goal (to eliminate y), and perform the task correctly to win the game. The txt2\pi algorithm uses the newly proposed FiLM^2 module and consists of integrating the visual input (grid world configuration) and the text input (descriptions) to learn a policy and baseline (actor-critic).

Pros
- The presentation is relatively clear.
- A new environment that can be impactful for the field.

Cons
- Certain design choices are not discussed properly.
- The experimental evidence is relatively weak.

(1) For RTFM, a complete list of possible elements would be helpful, at least in the supplementary. For example, the possible monsters/elements/weapons, etc.

(2) Some proposals lack motivation or explanation.
(2.1) How is Eq.(8) a summary?
(2.2) Why the attention in (18) should base on the vis-doc embedding instead of the goal embedding, or goal-conditioned doc embedding?
(2.3) It would be helpful to provide an example of the inventory description.
(2.4) What is the difference between the Goal and Goal-doc in Fig.3?

(3) Experiment
(3.1) "no group", "no dyna" and "no nl" seem to be suggesting a simpler environment. How is no natural language templated descriptions (no nl) considered as an easier scenario than with a template? The description would be harder to parse or understand without structure (template).
(3.2) It is mentioned several times that the models are "trained on one set of dynamics and evaluated on another set of dynamics". Specifically, what are the differences?
(3.3) For the curriculum training (Table 2), adding dyna does not hurt the performance (from 84 to 85). Further explanation would be helpful.
(3.4) How are the baselines and alternative methods perform in the full environment (+group etc.)?
(3.5) The texts on Fig.5 are not evenly spaced, which makes them harder to align with the attention (also some letters have dots above them). Does this figure correspond to (15)? In Fig.5b, a0 focuses on "gleaming" and a1 focuses on "lighting", which are not present in the entities (compared to the caption). Something is wrong or missing here.
(3.6) Eventually, the txt2\pi algorithm only achieves modest improvement over random agents (65% versus 50%), which is not very impressive.

## Update ##
Thank you for the thorough explanations, which are very helpful. It would be better to include them in the paper to clarify potential misunderstandings.

One more remark is that the goal-doc attention map (i.e., lg) in Fig. 5b focuses on lynx, which does not make sense in an environment with only a beetle and a wolf. This is strange and problematic given that it is an important attention map for many subsequent modules.

I have changed my score accordingly. It also needs to be considered that I'm not very familiar with the field.

**Experience Assessment:**

I do not know much about this area.

**Review Assessment: Checking Correctness Of Derivations And Theory:**

I assessed the sensibility of the derivations and theory.

**Review Assessment: Checking Correctness Of Experiments:**

I assessed the sensibility of the experiments.

**Review Assessment: Thoroughness In Paper Reading:**

I read the paper at least twice and used my best judgement in assessing the paper.

---

> ### Author Response · Authors · 2019-11-06
> **Author response**
>
> We thank the reviewer for their time and detailed feedback. We are happy they found the work interesting and with potential for impact in the field. We are confident that, following the points of clarification highlighted by the reviewer, we can resolve any ambiguities in the paper that might hinder such impact, and thank you for helping make the paper stronger as a result. We answer most of said points below. It would be extremely helpful to hear your thoughts on our responses, so that we can make the appropriate modifications to the paper. We emphasize that, given our paper length of under 8 pages, we can make such amendments while keeping the paper of comparative length to other submissions, and that you will be willing to consider revising your assessment in light of the clarifications once our discussion is over.
>
> We have made the following changes to the draft to account for your helpful feedback:
> Added a list of entities to Section G of the Appendix
> Added inventory text to each playthrough figure
> Vertically aligned attention x axis labels of attention maps such that they are more clear
>
> 1. Design choice
>
> Our modeling design choices mainly serve to build co-dependent features of environment observations, the goal, and dynamics encoded in language. Prior work use feature modulation (e.g. FiLM by Perez et. al) to incorporate goal features into a CNN that processes observations. In this work, we propose a method that supports feature modulation from observations to language as well as language to observations. Our work also attends to language at each layer of observation processing. We show that these design choices combine to provide significant gains over FiLM.
>
> 2. Experimental evidence
>
> We observe performance gains on two problems (RTFM and Rock-paper-scissors, the latter is found in Appendix E). For both problems, we show that RTFM consistently outperforms existing state-of-the-art methods, and that the performance across multiple trials.
>
> 3. For RTFM, a complete list of possible elements would be helpful, at least in the supplementary. For example, the possible monsters/elements/weapons, etc.
>
> Thank you for this suggestion. We added a list of possible elements in section G of the Appendix.
>
> 4. How is Eq.(8) a summary?
>
> Each layer of FiLM^2 encodes a higher-level representation of the inputs. Eq.(8) performs spatial max-pooling over this input. Hence it captures a spatial-invariant summary of the input at the (i)th layer.
>
> 5. Why the attention in (18) should base on the vis-doc embedding instead of the goal embedding, or goal-conditioned doc embedding?
>
> The per-layer attention in (18) is based on the vis-doc embedding instead of the goal because the goal is static given the episode. In contrast, the reasoning procedure requires that the model focus on different parts of the document during each reasoning stage (see reasoning procedure in introduction). To account for this, we allow each layer to attend to a different part of the document. We see in Table 1 that this is crucial to attaining good performance, and that the attention across different layers indeed pays attention to different parts of the document in Figure 5.
>
> 6. It would be helpful to provide an example of the inventory description.
>
> We added this to each playthrough figure in the paper. The inventory description consists of the name of the item that the agent possesses (e.g. “arcane hammer”).
>
> 7. What is the difference between the Goal and Goal-doc in Fig.3?
>
> Goal refers to the goal input (e.g. “defeat the Order of the Forest”). The two BiLSTMs (Goal-doc and Vis-doc) refer to two separate encodings of the document. The goal-doc encoding is subsequently used for attention with the goal. The vis-doc encoding is subsequently used for feature modulation and attention with the visual observations.
>
>
> 8. "no group", "no dyna" and "no nl" seem to be suggesting a simpler environment. How is no natural language templated descriptions (no nl) considered as an easier scenario than with a template? The description would be harder to parse or understand without structure (template).
>
> Yes, these are simpler variants of the game used for curriculum learning. “No nl” refers to a variant in which only one natural language template is used instead of sampling from all available natural language templates. In the former case, the model can assume one consistent pattern whereas in the latter case it cannot.
>
> [...continued in next post]

---

> > ### Author Response · Authors · 2019-11-06
> > **Author response pt. 2**
> >
> > 9. It is mentioned several times that the models are "trained on one set of dynamics and evaluated on another set of dynamics". Specifically, what are the differences?
> >
> > The dynamics refer to the rules of the game. Prior work study generalization across goals but not dynamics. In our case, given two games with the same goal (e.g. defeat the order of the forest) and characters (e.g. wolf and dragon), the dynamics of the game changes (e.g. which monster is on which team, which monster is weak to which element), hence the agent must dynamically formulate a strategy by reading to figure out which monster is on the correct team, and which item it needs to fight the element of that monster. Unlike prior work, given a goal such as “defeat the order of the forest”, the agent cannot memorise static associations such as “wolf is always on the order of the forest”, “wolf is always fire”, and “mysterious always beats fire” because in our case these dynamics are always changing and must be inferred through reading.
> >
> > 10. For the curriculum training (Table 2), adding dyna does not hurt the performance (from 84 to 85). Further explanation would be helpful.
> >
> > Dyna refers to the addition of moving monsters as opposed to static monsters. The random agent trained on 6x6+dyna achieves only 26% win-rate, which shows that handling moving monsters is difficult to learn. However once an agent has learned to handle static monsters (e.g. that it should not run into them before acquiring the correct item), its performance transfers to the scenario with moving monsters. The reason for this transfer is that the additional policy change the agent has to learn is to maintain a buffer distance between it and monsters, as opposed to maintaining a 1-cell distance between it and monsters. This delta is relatively small, which is why the performance transfers well.
> >
> > 11. How are the baselines and alternative methods perform in the full environment (+group etc.)?
> >
> > The baselines and alternatives do not perform well in the full environment. As seen in Figure 4, both conv and FiLM flatline at 20% win rate for the initial curriculum step, and achieve worse-than-20% performance on subsequent curriculum steps. We did not carry out ablation studies through the curriculum as doing so while computing variance is expensive. In practice, we find curriculum learning to be crucial, and that if a method fails in one stage of the curriculum (which the ablations do as seen in Figure 1), it is highly unlikely to perform well on the full environment.
> >
> > 12. The texts on Fig.5 are not evenly spaced, which makes them harder to align with the attention (also some letters have dots above them). Does this figure correspond to (15)? In Fig.5b, a0 focuses on "gleaming" and a1 focuses on "lighting", which are not present in the entities (compared to the caption). Something is wrong or missing here.
> >
> > We have adjusted the figures. The “dots” are periods separating sentences. The figure correspond to the per-layer vis-doc attention in (18). The goal-doc attention in (15) is denoted as “lg” in the figure. The caption refers to items present in the game, whereas the document refers to game rules. We do observe early-stage attention where the model attends to terms not relevant to solving the game, as your point out. However this occurs less frequently in late-stage attention (e.g. see a3, a4).
> >
> > 13. Eventually, the txt2\pi algorithm only achieves modest improvement over random agents (65% versus 50%), which is not very impressive.
> >
> > The randomly initialized agent’s performance is in the first row. For the full task, it is 23% and not 50%. In other words, our model obtains 65% compared to 23% on the full task.

---

### Official Review · AnonReviewer1 · 2019-10-24
**Official Blind Review #1**

**Rating:** 6

**Review:**

This paper constructs a new game that requires combining visual reasoning with text understanding to win.  The authors propose a new model txt2π, based on a new layer called FiLM², which combines text and visual features in a way that allows visual features to be encoded with knowledge of the text features (as in the FiLM layer from previous work), as well as text to be encoded with knowledge of the visual features. The model is trained to play the game using IMPALA.  Ablation studies show that the FiLM² layer leads to a substantial improvement, and also shows the necessity of curriculum learning.  Performance is still below human performance suggesting this is a promising area for future work.

The work here is complicated, but the writing and explanations are very clear. A new problem domain is introduced, which I think will be interesting to many other researchers.  I am not an expert in this area, so it's hard for me to give a confident assessment of the significance of this paper, so I give this paper a Weak Accept, despite enjoying the paper and having no specific points of criticism.

**Experience Assessment:**

I do not know much about this area.

**Review Assessment: Checking Correctness Of Derivations And Theory:**

I assessed the sensibility of the derivations and theory.

**Review Assessment: Checking Correctness Of Experiments:**

N/A

**Review Assessment: Thoroughness In Paper Reading:**

I made a quick assessment of this paper.

---

> ### Author Response · Authors · 2019-11-06
> **Author response**
>
> We thank the reviewer for their time and feedback. We appreciate it can be taxing to review papers in an area one is not an expert in, but we are happy to hear the paper was clear and interesting to the reviewer despite this. Regarding the score, if you believe you might be wrong in your assessment (which we understand if a concern), is it not sufficient to indicate this via the “Experience Assessment” field of the official review? Effectively penalising the paper by lowering your score as a way of indicating uncertainty without providing criticism we can respond to does not give us grounds for improving the paper. We ask that, if you honestly found the paper interesting impactful, and without obvious grounds for rejection based on you reading, that you consider raising your score to an accept. You are naturally welcome to further flag your uncertainty to the area chair if you feel the “Experience Assessment” field is not sufficient.

---

### Official Review · AnonReviewer4 · 2019-10-29
**Official Blind Review #4**

**Rating:** 6

**Review:**

First of all, I should acknowledge that this is a last-minute emergency review. Second, I should also acknowledge that I haven’t actively followed the specific research topic handled in this paper, although I work quite a lot in reinforcement learning and dialogue systems. The last paper I read on the topic was Branavan et al’s “Learning to win by reading manuals in monte-carlo framework” (ACL 2011).

This paper is about language-conditioned reinforcement learning, where the agent is required to perform machine reading of documents to learn policies to solve a task in new environments. The contribution of this paper is seemingly two fold: first, a new benchmark test suite is proposed (named “read to fight monsters” funny acronym), and second, an improved neural model (named “txt2pi”) which extends FiLM proposed for related tasks. Various experiments on txt2pi are conducted to show how the model performs against baselines, as well as it behaves under different learning settings.

Overall, this paper makes a good contribution to the research on deep models for language-conditioned reinforcement learning, but I see a number of things I miss in this paper.

The first one is about the test suite RTFM. The authors claim that it poses a new challenge that the agent has to understand the language specification of the goal, the environment dynamics, and the observations *all together* (compared to prior work that misses one or two), I am not confident that it is not challenging enough.

- The main concern I have is that they are all machine-generated, and soon or later, this kind of test suite could be beaten-to-death. A more ideal test suite would be the texts collected by humans, aiming for more diversity, ambiguity, and sometimes incorrectness.
- The RTFM would combinatorily generate a large number of environments, but still, they all look pretty simple. It would be interesting to have “partially specified” documents, such as “lightning monsters are known to be weak against grandmaster’s or soldier’s weapons, but it is not specifically known”. It would be interesting to have partially observable environments, where the agent has to do some information gathering in the environment to behave optimally.

The second one is about txt2pi model proposed in the paper.
- txt2pi adequately extends FiLM to handle multiple sources of information, but what is the lesson being learned?
I find the discussions in page 15 very enlightening. Could you move the content to the main text and defer the curriculum learning details to the appendix?

In summary, this paper proposes an interesting test suite for language-conditioned reinforcement learning, but I wish it is more realistic in the language data and more complex in the RL tasks.

---
I have updated my rating based on the reviewer response. Thank you for clarifying some of my concerns. I am still a bit reserved because of the use of synthetic corpus. However, at the same time, I think there could be a lot of follow-up papers based on this work.


**Experience Assessment:**

I have read many papers in this area.

**Review Assessment: Checking Correctness Of Derivations And Theory:**

I did not assess the derivations or theory.

**Review Assessment: Checking Correctness Of Experiments:**

I assessed the sensibility of the experiments.

**Review Assessment: Thoroughness In Paper Reading:**

I read the paper at least twice and used my best judgement in assessing the paper.

---

> ### Author Response · Authors · 2019-11-06
> **Author response**
>
> We thank the reviewer for their time and detailed feedback, especially given the short time-frame they presumably had to do this! We are happy to hear you find the work interesting and impactful, and understand your concerns. We respond to the points you make below, and hope through our discussion we can make suitable adjustments to the paper’s presentation to satisfy you that the current work at least proves the concept enough to warrant publication, and that you will consider revising your assessment.
>
> We emphasise that, at a length of under 8 pages, we have ample room to expand on the points discussed below in the paper (e.g. by moving content from the appendix as you suggest) while maintaining a paper length in line with most other submissions.
>
> We have made the following changes to the draft to account for your helpful feedback:
> - Added findings from page 15 (the rock-paper-scissors experiments) to Section 5.1 of the main text
>
> 1. Difficulty of RTFM
>
> While future work can and should focus on natural language, this paper aims to prove the concept of generalization via reading in a controlled environment. To illustrate this, let's reiterate on the Civ paper that you gave as an example above. While that paper uses real language (the Civ manual) it only investigates shallow conditioning on static text through hand crafted features and heuristics for grounding. We found that if only conditioning on a static document describing dynamics, the agent is not encouraged to actually perform reading comprehension and instead memorises the single game it was trained on. Only by procedurally generating the environment dynamics as well as the text describing these can we test for generalization of reading comprehension and grounding capabilities of the RL agents.
>
> Moreover, RTFM is complex enough to challenge state-of-the-art methods such as IMPALA and FiLM. These methods solve environments such as DMLab, Atari (https://arxiv.org/pdf/1802.01561.pdf), and GridLU-relations (https://arxiv.org/pdf/1806.01946.pdf) but struggle on RTFM. With our methods, we find it is difficult to make progress without curriculum learning, which ideal methods should not require.
>
> We like your suggestion on partial specifications, and will study it in future work.
>
> 2: Lessons learned from txt2pi
>
> The key lesson learned here is that when modeling complex textual inputs, such as dynamics that necessarily change the reasoning steps, we need to not only model how textual features modify representations of observations, but how observations modify representations of text. In this work we proposed two such changes, one is the feature modulation of text given observations in the FiLM^2 layer, the other is the layer-wise vis-doc attention over the document given the layer’s summary such that the model can attend to different parts of the document in each reasoning stage.
>
> We added the findings from page 15 into section 5.1 of the main text. Thank you for reading our appendix :)

---

### Author Response · Authors · 2019-09-26
**Errata**

Apologies to the readers - we identified an error in Figure 4. The blue lines should be "txt2pi" instead of "film^2".

---

### Decision · Program_Chairs · 2019-12-19

**Decision:**

Accept (Poster)

**Comment:**

This paper proposes RTFM, a new model in the field of language-conditioned policy learning. This approach is promising and important in reinforcement learning because of the difficulty to learn policies in new environments.

Reviewers appreciate the importance of the problem and the effective approach. After the author response which addressed some of the major concerns, reviewers feel more positive about the paper. They comment, though, that presentation could be clearer, and the limitations of using synthetic data should be discussed in depth.

I thank the authors for submitting this paper.